# The reversal in the cryptocurrency market before and during the Covid-19 pandemic: Does investor attention matter?

**Huy Pham[1], Trang Ngoc Doan Tran[2], Ngoc Thi Thanh Nguyen[2], Khoa Dang Duong [2]\***

**1** The Business School, RMIT University Vietnam, Ho Chi Minh City, Vietnam, **2** Faculty of Finance and Banking, Ton Duc Thang University, Ho Chi Minh City, Vietnam

\* duongdangkhoa@tdtu.edu.vn

**Data Availability Statement:** The data that support the findings of this study are openly available in

## Abstract

This study delves into the impact of reversals and investor attention on cryptocurrency returns before and during the COVID-19 pandemic. We employ the Two Stages Least Squares to analyze a sample of the top 20 cryptocurrencies from January 2016 to April 2021. Our results reveal that investor attention positively influences bitcoin returns in both periods, with a more pronounced effect during the pandemic. Conversely, reversals demonstrate a positive correlation with cryptocurrency returns before the outbreak but a negative relationship during the pandemic. Our robustness test further indicates that investor attention positively affects the returns of small and medium-cap cryptocurrencies, while reversals only exhibit positive consequences for small-cap cryptocurrencies. Additionally, our findings highlight stablecoins as a safe haven during the epidemic. The results suggest that investor attention has little influence on the returns of stablecoins, indicating that these coins are primarily resistant to market sentiment due to their inherent stability. The negative impact of the pandemic on the crypto market demonstrates a downward trend through each wave. Despite aligning with attention-induced price pressure and behavioral finance hypotheses, our results do not support efficient market theory or the notion of heterogeneity among investors. This research provides valuable insights for investors and policymakers in devising effective strategies for the cryptocurrency market.

## Introduction

Recently, cryptocurrencies have been developing rapidly and attracting the attention of academics and practitioners. Cryptocurrencies are attractive because the blockchain platform allows individuals to conduct transactions without financial institutions while ensuring safety and speed [1]. However, cryptocurrencies are more complex than traditional equity assets, so traditional asset pricing models and risk factors fail to predict crypto returns [2]. The cause stems from needing more popular fundamental information such as dividends and book value. Therefore, external factors such as irrational noise trading, volatility, and investor attention become efficient predictors of crypto returns.

Harvard Dataverse at https://doi.org/10.7910/DVN/2W5ZNB.

**Funding:** This study is support by Ton Duc Thang University and RMIT University Vietnam.

**Competing interests:** The authors have declared that no competing interests exist.

Recently, the reversals in the cryptocurrency market have become an essential topic for the following reasons. Firstly, identifying reversal patterns can help traders decide when to enter or exit positions. Recognizing early signs of a market reversal allows investors to capitalize on potential price changes, maximizing profit or minimizing losses. Secondly, Reversals can signal shifts in market sentiment and trends. By studying these patterns, investors can better manage risks associated with their investments. Understanding potential reversals allows for implementing risk mitigation strategies, such as setting stop-loss orders or adjusting portfolio allocations. Finally, researching reversals contributes to a deeper understanding of market dynamics. This knowledge is crucial for developing effective trading strategies. Whether investors are short-term traders or long-term investors, being aware of potential reversals enhances their ability to adapt their strategies based on changing market conditions.

Depending on different market cycles, retail investor attention can cause volatility because it initiates buying and selling behaviors in the market [3]. Renault [4] argues that investor attention can raise expectations and overreact, leading to a reversal effect. Heyman et al. [5] show that increasing Google search volume may also represent investor overreaction to supposedly profitable signals. This action leads to a significant increase in returns from reversal. The reversal effect mainly occurs during the volatility period, and the market overreacts to the news. Kozlowski et al. [6] show that reversals exist during both bull and bear markets. Excessive optimism or pessimism can cause a price shock. This shock leads to a temporary imbalance in supply and demand that causes a short-term reversal. Therefore, it is worth testing the simultaneous relationship between investor sentiment revealed by search and the reversal effect on crypto returns.

Several incentives drive us to carry out this investigation. Prior research [7–9] has shown that reversals may impact investors' earnings due to psychological factors. The overreaction theory posits that investors may respond excessively to short-term news or events, resulting in prices above their long-term average. However, research on identifying a meaningful correlation between reversals and investor attention is still uncommon. Furthermore, the correlation between these two parameters has yet to be fully used. Shen et al. [10] only include reversal as the third variable in the CAPM model inside the cryptocurrency market. Kozlowski et al. [6] demonstrate that the reversal effect is present in high- and low-market volatility conditions. Nevertheless, their conclusions continue to be inconclusive and subject to debate. Previous studies also have not highlighted the relationship between the reversal of crypto returns and investor attention during the epidemic. Hence, this research examines the intricate correlation between reversal and investor attention, particularly in the pandemic era, to provide a thorough understanding of their connection.

Furthermore, due to the high level of instability during the pandemic, our objective is to examine the ability of cryptocurrencies to serve as safe-haven assets throughout the epidemic. This analysis will provide investors with a secure viewpoint for their investments. According to Conlon et al. [11], digital assets like cryptocurrencies have safe-haven characteristics similar to conventional precious assets like gold during the pandemic. Conversely, Güler [12] demonstrates that Bitcoin may be a safeguard to mitigate portfolio risk during the epidemic. The primary reason for this phenomenon may be the emotional and speculative influence and the atypical expansion of the cryptocurrency market. Hence, we use the verified COVID-19 cases and confirmed COVID-19 fatality data in the United States to examine the influence of pandemic news on the characteristics of crypto-safe havens.

We follow Subramaniam and Chakraborty [3] to estimate investor attention using Google trends. We also follow Kozlowski et al. [6] to estimate reversal as the previous day's returns. We collected a sample of 6,606 daily observations of the top 20 largest cap cryptos from January 2nd, 2016, to April 15th, 2021. We utilize the 2SLS regression method to estimate the main

findings and robustness tests. Our primary findings suggest that reversal positively affects returns before COVID-19 pandemic. However, the reversal negatively affects crypto returns during the pandemic. It could be because market inefficiencies theory leads to a higher information asymmetry during the epidemic, so investors overreact to market volatility.

Our findings indicate that investor attention positively empowers crypto returns before and during the pandemic. The impact of investor attention on returns during the pandemic is 1.09% stronger than before. The results show that investors pay greater attention to the crypto market during COVID-19, pushing higher returns. At the same time, the increase in COVID-19 cases and deaths also positively increases crypto returns, demonstrating their safe-haven function. Our findings align with Subramaniam and Chakraborty [3], Ozdamar et al. [7], Conlon et al. [11], Güler [12], and the attention-induced price pressure hypothesis of Barber and Odean [13].

Moreover, our findings report that investor attention positively affects small and medium-cap crypto returns, while reversal positively affects small-cap crypto returns. At the same time, the stability-specific subsamples show that investor attention does not affect stable coin returns. In addition, reversal positively impacts stable coin returns during the pandemic, indicating their stable and safe haven properties. During the epidemic, some capital flows into stable coins to hedge risk. Our findings also show that the epidemic's negative impact on crypto returns decreases in each Covid wave. While the negative effect of reversal on returns decreases gradually through each pandemic wave, investor attention favorably affects the crypto returns in the third wave.

Our paper significantly contributes to the expanding body of literature on cryptocurrency topics. Notably, we are among the first to explore the simultaneous influence of investor attention and reversal on crypto returns before and during the pandemic. We adopt a methodology for calculating investor attention directly, following Subramaniam and Chakraborty [3] and contrasting it with adjustments made by Da et al. [14], Duong et al. [15]. Moreover, we enrich our findings by examining a broader spectrum of cryptocurrencies, moving beyond those with prominent recognition. This approach contributes to a more comprehensive understanding of the impact of investor sentiment on company earnings [16]. Building on the work of Bouteska et al. [17] regarding the influence of investor sentiment on Bitcoin's performance during the pandemic, we aim to delve deeper into the effects of investment sentiment on the top 20 cryptocurrencies in the global market throughout the COVID-19 epidemic. Finally, we use the panel unit root test instead of the original time-series unit root test. Compared to time-series unit root tests, panel unit root tests can be much more powerful when looking at many cryptos, especially in small samples [18].

## Literature review

### The safe-haven properties of cryptocurrency

Li and Miu [19], Li et al. [20] assert that a hedging asset is characterized by a zero or potentially negative correlation with another asset. This type of asset earns the designation of a safe-haven asset when the zero or negative correlation persists, particularly amid financial crises. Jana and Sahu [21] discover that stablecoins such as Tether and USD Coin function as effective hedges in regular economic conditions and serve as safe havens during periods of economic decline. Ugolini et al. [22] demonstrate that cryptocurrency adoption by equity investors can serve as a strategy to hedge against declines in stock prices. Jana and Sahu [23] also point out a negative conditional correlation between equities and cryptocurrencies before a crisis, turning positive during the crisis, except for Tether. Significantly, Jana et al. [24] also find that Tether functions as a safe-haven asset during periods of financial turmoil. BenSaïda [25] shows that digital currencies emerge as safe havens for nearly all fiat currencies.

Furthermore, Jana and Sahu [26] affirm that in a stable economic climate, cryptocurrencies exhibit no significant connection with the Indian stock market. However, in times of financial upheaval, Bitcoin, Ethereum, and Cardano demonstrate a positive correlation with the stock market. The study identifies Bitcoin, Ethereum, Dogecoin, and Cardano as viable options for diversification and hedging in normal economic conditions. Nevertheless, only Dogecoin may function as a safe-haven asset during financial stress.

## Reversal and returns

Ozdamar et al. [7], Hung and Yang [8] contribute significantly to the literature on behavioral finance, revealing that reversals positively impact stock returns. This finding aligns with the overreaction hypothesis, which suggests that investors may overreact to short-term news or events, causing prices to overshoot their long-term average. The subsequent reversal can thus present profitable investment opportunities. Hung and Yang [8] studied US shares between 1973 and 2015, and they report that the MAX effect offers evidence that certain lottery stocks with limits to arbitrage can experience significant positive returns due to the reversal effect. Ozdamar et al. [7] investigated the link between the reversal effect and stock returns in the global Islamic equity market, offering valuable insights into the role of behavioral factors in shaping investor behavior and market outcomes. These findings highlight the potential relevance of the overreaction hypothesis in predicting stock returns and making sound investment choices.

Kozlowski et al. [6], Ozdamar et al. [7], Eom and Park [9], Bali et al. [27] have indicated evidence of a negative reversal effect on stock returns. These findings contrast the results of other studies and align with the theory of market inefficiencies, particularly the underreaction hypothesis. The underreaction hypothesis is a theory in finance that suggests that investors tend to under-react to newly available information and news announcements, leading to a delayed adjustment of stock prices. According to this hypothesis, investors may only partially process all the relevant information about a stock immediately and instead require time to fully incorporate information into the market price. Specifically, Bali et al. [27] suggest that the negative effect of Reversal on US stock returns is due to the inherent characteristics of small and illiquid stocks. Meanwhile, Eom and Park [9] report a negative reversal effect in the US, Japanese, Chinese, and Korean markets, which is consistent with the post-earnings announcement drift (PEAD) phenomenon, which suggests that investors tend to under-react to earnings surprises and cause a gradual adjustment of stock prices. Kozlowski et al. [6] report a negative impact of a reversal on cryptocurrency returns from 2015 to 2019, consistent with the underreaction hypothesis. This hypothesis suggests that investors tend to under-react to newly available information, leading to a price reversal in the subsequent period.

However, Cao et al. [28], Duong et al. [29] indicate that the reversal effect does not significantly influence stock returns, which is consistent with the efficient market theory, which asserts that the market is highly efficient and that stock prices reflect all publicly available information. In particular, Cao et al. [28] examined the relationship between daily returns reversal and stock returns in the US between March 2003 and December 2016. They concluded that the former did not significantly affect the latter. Similarly, Duong et al. [29] investigated the impact of reversal on stock returns in the Taiwan market from January 2005 to March 2021. They found insignificant reversal effects, supporting the idea that efficient markets reflect all available information.

To summarize the literature on reversal, there have been mixed findings on its effect on returns. The positive effect of reversal on returns is consistent with the overreaction hypothesis in behavioral finance. At the same time, the negative impact aligns with the underreaction

theory. Considering these theories, we propose our first hypothesis to test the specific impact of a reversal on returns.

**Hypothesis 1:** *Reversal significantly impacts crypto returns before and during the COVID-19 pandemic.*

## Investor attention and returns

Subramaniam and Chakraborty [3], Da et al. [14] suggest that an increase in the search volume index (SVI) is associated with higher asset returns. These findings support the attention-induced price pressure hypothesis [13]. This hypothesis posits that investors tend to focus on stocks that receive more attention, leading to increased buying pressure and, consequently, higher prices. This attention effect can be observed in various forms, such as media coverage, internet searches, or analyst recommendations. Da et al. [14] suggest that a higher search volume for a stock is positively associated with its following-day returns. They attributed this effect to increased investor attention, which leads to more informed trading decisions and improves market efficiency. Subramaniam and Chakraborty [3] also studied the relationship between social media sentiment and stock returns. They found that higher sentiment intensity, as measured by SVI, leads to positive abnormal returns, consistent with the attention-induced price pressure hypothesis. These studies provide empirical evidence for the importance of investor attention in forming asset prices and the potential effects of information cascades on market outcomes.

Da et al. [30], Chen et al. [31] employed negative search keywords to gauge investor sentiment. They determined that it reduces financial asset returns. These findings support the heterogeneity of investors theory, which contends that investors may react differently to market events based on risk tolerance and investment objectives. Some investors may be risk-averse and respond with apprehension during market turbulence. In contrast, others may be more risk-tolerant and view market downturns as an opportunity for investment. Da et al. [30] developed the FEARS index to gauge investor fear using 30 negative financial keywords in the US between 2004 and 2008. Increasing investor interest in these keywords corresponds to decreased stock returns. Chen et al. [31] extended this concept to the crypto market by devising an investor fear index based on COVID-19-related keywords. Their study also found that heightened fear sentiment contributes to an erratic market and reduces returns for Bitcoin. These findings highlight the significance of investor sentiment and perception in shaping market outcomes and emphasize the significance of tracking investor sentiment through search volume or social media for making sound investment choices.

Urquhart [32], Figà-Talamanca and Patacca [33] contribute to the literature by suggesting that the relationship between the search volume index (SVI) and Bitcoin returns may not always be statistically significant, aligning with the efficient market hypothesis (EMH). Urquhart [32] employed the autoregressive vector method to analyze Bitcoin returns from August 2010 to July 2017 and concluded that SVI did not significantly impact returns and realized volatility in all statistical time series. Similarly, Figà-Talamanca and Patacca [33] employed a multivariate VAR-EGARCH model to investigate the impact of SVI and volatility on Bitcoin returns, yielding results that did not reflect a significant association between SVI or volatility and Bitcoin returns. Investors are typically drawn towards Bitcoin following price and information shocks.

Similarly, the aim is to assess the impact of investor attention on crypto returns. We put forward the second hypothesis as follows:

**Hypothesis 2:** *Investor attention affects crypto returns before and during the pandemic.*

## Other determinants of cryptocurrency returns

Size is a proxy for common risk in financial assets. Vo [34] show that size positively correlates with Vietnam stock returns from 2007 to 2012. On the other hand, Li et al. [35] also report that the size negatively affects the returns in the sample of more than 1800 cryptocurrencies from January 2014 to May 2019. Furthermore, Ozdamar et al. [7] show that market value is not statistically significant with the returns of 523 cryptocurrencies from January 2014 to September 2020.

Several studies show different effects of turnover on the returns of financial assets. Dash and Maitra [36] document that turnover positively affects stock returns in Indian markets from 2002 through 2014. On the other hand, Hung and Yang [8], Duong et al. [29] show an inverse relationship between turnover and stock returns in Taiwan. Kozlowski et al. [6] show that turnover is not statistically significant between turnover and returns in a sample of 200 cryptocurrencies from 2015 to 2019.

Volatility and Skewness are risk factors that affect the returns on financial assets. Fang et al. [37] document that an increase in total volatility leads to an increase in stock returns in the Shanghai stock market from June 1999 to June 2019. In contrast, Lee [38] shows that total volatility has an inverse effect on stock returns with 1295 NYSE/AMEX stocks from July 1967 to December 2007. Foroutan and Lahmiri [39] found no significant change in the relationship between returns and volatility during the pandemic in cryptocurrency markets with the top 20 cryptocurrencies listed on CoinMarketCap, covering the period from January 1st, 2020, to December 31st, 2020. About Skewness, Fang et al. [37] show that total Skewness positively affects stock returns. In addition, Ozdamar et al. [7] indicate that TSKEW is not statistically significant with the returns.

## Covid-19 and returns

Global economic and financial progress faces significant disruptions due to the COVID-19 pandemic [40]. In response, international investors are redirecting their focus toward more reliable assets to safeguard their portfolios, with cryptocurrencies emerging as a notable choice since they serve as safe havens for nearly all fiat currencies [25]. However, the pandemic has introduced challenges for Bitcoin and the cryptocurrency markets, marked by heightened volatility [41]. This unprecedented global health crisis has elevated concerns in the financial landscape and various financial markets. Recognizing the economic challenges posed by the pandemic, numerous studies have sought to investigate its impact on cryptocurrency markets. Sahoo [42] conducts an empirical examination of the influence of the COVID-19 pandemic on cryptocurrency market returns, confirming a unidirectional causal relationship between COVID-19 confirmed cases and death cases to cryptocurrency returns. Sahoo and Rath [41] delve into the causal connection between the COVID-19 pandemic and Bitcoin returns, employing a time and frequency domain Granger causality framework. They discover a causal relationship between COVID-19 and the fluctuations in Bitcoin returns over time. Their findings also reveal that the impact of COVID-19 on Bitcoin returns differs across various time intervals, including short, medium, and long-term.

## Data and methodology

### Cryptocurrency data

We use a sample that includes daily data for the top 20 cryptos by market cap. Our sample represents over 80% of the total Cryptocurrency market value on 1st Arp 2021. Daily trading data is from coinmarketcap.com. At the same time, we collect data on COVID-19 cases and deaths

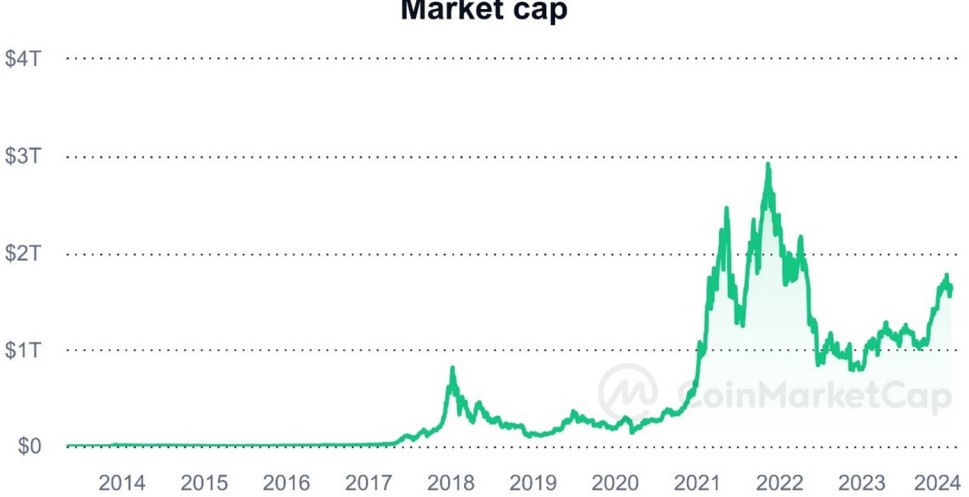

**Fig 1. Cryptocurrency market capitalization from 2016 to 2021 (source: CoinMarketCap).**

in the US from ourworldindata.org. We follow Duong et al. [43] to winsorize all variables at 5% and 95% to remove outliers. The final sample includes 6,606 daily observed data from January 2nd, 2016, to April 15th, 2021. We select the sampling period from January 2016 to April 2021 for several reasons. Fig 1 shows the high volatility of cryptocurrency market capitalization between 2016 and 2021. Therefore, we aim to investigate the asset bubble issues when various asset classes experience significant overvaluation relative to their fundamental economic value [44]. This period offered an opportune context to explore investor reactions and assess how investor psychology influenced the profitability of virtual currencies. Additionally, our focus extended to examining the influence of investor sentiment on reversals, particularly during the pandemic, to gain insights into the dynamics of market behavior during periods of heightened uncertainty and volatility. S1 Appendix shows detailed market capitalization and market share of the top twenty cryptocurrencies.

## The measure of reversal

Our dependent variable is the daily return of cryptocurrency i on day t, which is defined as:

$$R_{i,t} = \ln(P_{i,t}/P_{i,t-1})*100 \tag{Eq 1}$$

We then follow Kozlowski et al. [6] to conduct lag daily return to measure the reversal (REV) in day t, which is the return of the cryptocurrency i over day t-1.

## The measure of Google Search Volume Index (SVI)

We collect data on the search frequency of cryptocurrencies on Google Trends. Google standardizes the search volume index (SVI) from zero to one hundred. At the same time, Google scales SVI as a percentage of search topics in a particular geographic unit overall relevant search topic [3]. We examine the US market to assess the specific impact of COVID-19 on the US. At the same time, the search keywords are the cryptocurrency names classified under the financial category to avoid duplicate or irrelevant search results. We collect keyword data related to the top 20 cryptocurrencies on CoinMarketCap every three months to take daily frequency data on Google trends. The daily changes in SVI data were derived by excluding the initial data point for each of the 3-month windows.

We follow Subramaniam and Chakraborty [3] to calculate investor attention directly without any adjustment, as in Da et al. [14], Duong et al. [15]. The daily log change of SVI on day t is as follows:

$$\Delta SVI_t = \ln(SVI_t) - \ln(SVI_{t-1}) \tag{Eq 2}$$

## Estimation models

We follow Subramaniam and Chakraborty [3], Kozlowski et al. [6] to construct models to examine the effect of reversal and investor attention on crypto returns. We also add Size, Turnover, Volatility, and Skewness as our control variables in each regression model. Model (1) is defined in the pre-COVID-19 period. Where $R_{i,t}$ is the return of cryptocurrency i on day t.

$$R_{i,t} = \beta_0 + \beta_1 \text{Delta SVI}_{i,t} + \beta_2 \text{REV}_{i,t} + \sum \gamma_j \text{Controls}_{i,t} + \varepsilon_{i,t} \tag{1}$$

Sahoo and Rath [41], Sahoo [42] empirically investigate the impact of the COVID-19 pandemic on cryptocurrency market returns and reveal a causal relationship between COVID-19 and the price returns of cryptocurrency. Therefore, to assess the impact of the COVID-19 epidemic, we follow Baig et al. [45] to conduct further assessment of the number of confirmed COVID-19 cases (WCC) and COVID-19 deaths (WCD) in the US. Models (2) and (3) are built to examine the impact of WCC and WCD during COVID-19, respectively. The definition of COVID-19 began on January 22nd, 2021 when the US recorded its first COVID-19 case. Models (2) and (3) are as follows:

$$R_{i,t} = \beta_0 + \beta_1 \text{Delta SVI}_{i,t} + \beta_2 \text{REV}_{i,t} + \beta_3 \text{WCC}_{i,t} + \sum \gamma_j \text{Controls}_{i,t} + \varepsilon_{i,t} \tag{2}$$

$$R_{i,t} = \beta_0 + \beta_1 \text{Delta SVI}_{i,t} + \beta_2 \text{REV}_{i,t} + \beta_3 \text{WCD}_{i,t} + \sum \gamma_j \text{Controls}_{i,t} + \varepsilon_{i,t} \tag{3}$$

We follow Yousaf et al. [46] to define the dummy variable for the COVID-19 period. The dummy variable is one during the pandemic (from January 22nd, 2021, to April 15th, 2021) and 0 for the rest. We construct the interaction variables by multiplying them by a dummy to see the most affected factor in this period. Models (4) and (5) show the impact of WCC and WCD on the model, respectively:

$$R_{i,t} = \beta_0 + \beta_1 \text{Delta SVI}_{i,t} + \beta_2 \text{REV}_{i,t} + \beta_3 \text{LnWCC}_{i,t} + \beta_4 \text{DeltaSVI\_Dummy}_{i,t}$$
$$+ \beta_5 \text{TURN\_Dummy}_{i,t} + \beta_6 \text{REV\_Dummy}_{i,t} + \sum \gamma_j \text{Controls}_{i,t} + \varepsilon_{i,t} \tag{4}$$

$$R_{i,t} = \beta_0 + \beta_1 \text{Delta SVI}_{i,t} + \beta_2 \text{REV}_{i,t} + \beta_3 \text{LnWCD}_{i,t} + \beta_4 \text{DeltaSVI\_Dummy}_{i,t}$$
$$+ \beta_5 \text{TURN\_Dummy}_{i,t} + \beta_6 \text{REV\_Dummy}_{i,t} + \sum \gamma j \text{Controls}_{i,t} + \varepsilon_{i,t} \tag{5}$$

Where $R_{i,t}$ is the return on cryptocurrency i on day t. Delta $SVI_{i,t}$ denotes the daily log change of SVI; $REV_{i,t}$ denotes reversal; $LnWCC_{i,t}$ denotes confirmed COVID-19 cases; $LnWCD_{i,t}$ denotes COVID-19 deaths; $Controls_{i,t}$ includes turnover, size, total volatility, total skewness; and $\varepsilon_{i,t}$ is a random error. Detailed definitions for each variable can be found in S2 Appendix.

## Methodology

Initially, we employed the Hausman and Redundant Tests to choose between the Pooled Ordinary Least Square (OLS), Fixed Effects Model (FEM), and Random Effects Model (REM). By

using these tests, we can ensure the validity and reliability of our chosen estimation method. Ordinary least-squares (OLS) models operate because the analysis constructs a model representing the relationship between one or more explanatory variables and a continuous, or at least interval, outcome variable. This model aims to minimize the sum of squared errors, where an error is defined as the disparity between the actual and predicted values of the outcome variable [47]. Fixed-effects estimation focuses on individuals with multiple observations and calculates effects exclusively for variables that exhibit variation across these observations. This approach assumes that the impact of unchanging, unmeasured variables can be represented by incorporating time-invariant individual-specific dummy variables. In contrast, a random effects model is a statistical model in which the model parameters are treated as random variables. Random effects are typically assigned to predictors whose observed levels in the study represent a sample drawn from a larger population of levels. Designating a variable as a random effect is appropriate when the goal is to generalize findings to the broader population of potential levels for that variable [48].

However, Duong et al. [43] argue that OLS, FEM, and REM are susceptible to violating endogeneity and heterogeneity issues. Due to the inherent limitations of OLS in estimation, these methods can generate biases due to autocorrelation and endogeneity issues. Consequently, the Wald test is employed to investigate the presence of heteroskedasticity. If the results violate heteroskedasticity, we follow Entrop et al. [49], Li et al. [50] to apply Two-Stage Least Squares (2SLS) regression to address these endogeneity issues by using instrumental variables to replace the endogenous variables in the regression model. 2SLS regression analysis is a statistical method employed in analyzing structural models, representing an extension of the Ordinary Least Squares (OLS) approach. This technique is beneficial when there is a correlation between the dependent variable's error terms and the model's independent variables. Addressing endogeneity concerns through a two-step process, where the first stage calculates predicted values of endogenous variables using the instrumental variables. These predicted values are then used to estimate the second-stage regression model. 2SLS helps mitigate biases in parameter estimates that may arise in such correlation, making it a valuable tool for modeling situations with potential endogeneity issues [51]. Therefore, 2SLS provides better and more accurate estimations than conventional panel regression methods.

We also analyze the subsample of the sample according to market cap, stability, and COVID-19 waves to clarify the crypto characteristics further. We adopt a robustness test methodology proposed by Subramaniam and Chakraborty [3], Hung and Yang [8] to enhance our comprehension of the influence of investor attention and reversals on cryptocurrency returns across distinct sub-samples categorized by market capitalization. Additionally, we employ the approach presented by Wang et al. [52] to divide the sample into stable and traditional coins. To delve into the nuanced impact of the COVID-19 pandemic on cryptocurrencies, we employed John Drake's methodology, enabling us to scrutinize the effects of each pandemic wave on crypto returns in a detailed manner.

## Empirical findings and discussion

### Descriptive statistics

Table 1 demonstrates that the average daily crypto returns are 0.852% per day. At the same time, the average size is 23.224. The mean Delta SVI is 0.005; the standard deviation is 0.6, with the five percentile at -1.035 and the 95 percentile at 1.070. This result is similar to Subramaniam and Chakraborty [3]. In contrast, the average reversal is 0.952%, with a standard deviation of 9.087%. The median is 0.137%, a five percentile of -8.759%, and a 95 percentile of

**Table 1. Descriptive statistics.**

| Variable | N | Mean | Std. | Median | 5th | 95th |
|---|---|---|---|---|---|---|
| Ret (%) | 6,606 | 0.852 | 8.852 | 0.093 | -8.937 | 12.112 |
| SZ | 6,606 | 23.224 | 2.030 | 23.213 | 19.437 | 26.036 |
| Delta SVI | 6,606 | 0.005 | 0.600 | 0.000 | -1.035 | 1.070 |
| REV (%) | 6,606 | 0.952 | 9.087 | 0.137 | -8.759 | 12.720 |
| TURN (%) | 6,606 | 21.655 | 31.572 | 10.359 | 0.958 | 80.823 |
| TVOL | 6,606 | 6.030 | 5.173 | 4.839 | 1.420 | 13.414 |
| TSKEW | 6,606 | 0.571 | 1.593 | 0.474 | -2.188 | 2.901 |

Notes: The table reports descriptive statistics. All variable definitions are reported in S2 Appendix.

12.720%. The average turnover is 21.655% per day. In addition, the mean values of TVOL and TSKEW are 6,030 and 0.571, respectively.

## Pearson correlation matrix

Table 2 reports the Pearson correlation matrix between six variables. We also test the correlation between the Delta SVI, REV, TURN, SZ, TVOL, and TSKEW. We find that reversal has a negative relationship with size. The correlation coefficient of reversal with the size is -0.031. This result is similar to Kozlowski et al. [6] study that reversal tends to be more assertive with small-cap cryptocurrencies.

Moreover, Delta SVI positively correlates with turnover. It is consistent with Urquhart [32]. In addition, we also find that multicollinearity does not occur in our sample because the VIF test shows that the average VIF is only 1.21 [43].

## Panel unit root test

Models (1), (2), (3), (4), and (5) may yield misleading outcomes if the involved variables lack stationarity. A stationary variable is characterized by a consistent mean, variance, and autocorrelation structure over time without periodic fluctuations. Developing theories and models become considerably more intricate when dealing with non-stationary variables. Therefore, it is imperative to incorporate stationary variables in our models. In line with this, we follow

**Table 2. Pearson correlation matrix.**

| | Delta SVI | REV | TURN | SZ | TVOL | TSKEW | VIF |
|---|---|---|---|---|---|---|---|
| **Delta SVI** | 1 | 0.03101*** | 0.026** | -0.005 | 0.006 | 0.003 | 1 |
| | | (0.0119) | (0.0322) | (0.6865) | (0.6539) | (0.7887) | |
| **REV** | | 1 | 0.1399*** | -0.03094*** | 0.20361*** | 0.10505*** | 1.06 |
| | | | (<0.0001) | (<0.0001) | (<0.0001) | (<0.0001) | |
| **TURN** | | | 1 | -0.18309*** | 0.13356*** | -0.06715*** | 1.11 |
| | | | | (<0.0001) | (<0.0001) | (<0.0001) | |
| **SZ** | | | | 1 | -0.17221*** | -0.33163 *** | 1.18 |
| | | | | | (<0.0001) | (<0.0001) | |
| **TVOL** | | | | | 1 | 0.49539*** | 1.41 |
| | | | | | | (<0.0001) | |
| **TSKEW** | | | | | | 1 | 1.52 |

Notes: The table shows a Pearson correlation matrix for independent variables. All variable definitions are reported in S2 Appendix.

**Table 3. Panel unit root test.**

| LEVEL | LLC | IPS | ADF | PP |
|---|---|---|---|---|
| Ret | -86.3099 | -52.0774 | 1689.60 | 1988.22 |
|  | (<0.0001) | (<0.0001) | (<0.0001) | (<0.0001) |
| Delta SVI | -41.5737 | -44.1604 | 1440.65 | 1249.06 |
|  | (<0.0001) | (<0.0001) | (<0.0001) | (<0.0001) |
| REV | -76.3307 | -51.9526 | 1667.98 | 1991.67 |
|  | (<0.0001) | (<0.0001) | (<0.0001) | (<0.0001) |
| TURN | -6.85708 | -9.35110 | 211.075 | 684.016 |
|  | (<0.0001) | (<0.0001) | (<0.0001) | (<0.0001) |
| SZ | -23.9098 | -62.3497 | 1828.26 | 838.939 |
|  | (<0.0001) | (<0.0001) | (<0.0001) | (<0.0001) |
| TVOL | -8.73879 | -33.6607 | 1162.59 | 2056.98 |
|  | (<0.0001) | (<0.0001) | (<0.0001) | (<0.0001) |
| TSKEW | 2.85444 | -33.8508 | 1121.16 | 1970.58 |
|  | (<0.0001) | (<0.0001) | (<0.0001) | (<0.0001) |
| LnWCC | 676.287 | -15.0926 | 338.002 | 1740.69 |
|  | (<0.0001) | (<0.0001) | (<0.0001) | (<0.0001) |
| LnWCD | 466.254 | -16.6039 | 417.090 | 1781.92 |
|  | (<0.0001) | (<0.0001) | (<0.0001) | (<0.0001) |

Note: Table 3 shows panel unit root in level series, including individual effects and individual linear trends in the test model, automatic lag length selection based on SIC from 1 to 3. The value in parentheses is the significance level.

Subramaniam and Chakraborty [3] to employ the panel unit root test using the methodologies of Levin et al. [53], Im et al. [54]—IPS, ADF—Fisher, and P.P.—Fisher. Our findings indicate that all four methods reject the null hypothesis (series is non-stationary), suggesting that the variables are stationary.

Table 3 reports that all variables exhibit stationarity in their level series across the four tests, as indicated by p-values below the 5% significance level (except TSKEW, lnWCC, and lnWCD are insignificant at the LLC method). Hence, it is possible to construct the model in the level series.

## Estimation results and discussion

Table 4 presents the results of the fixed effect model (FEM) regression because the results of the Hausman test with a P-value less than 0.001 show that FEM is more suitable than REM. The Redundant Fixed Effects Test suggests that FEM is more suitable than OLS in the model. However, the Wald test indicates a heteroskedasticity issue with FEM estimations. Therefore, we employ the 2SLS regression method to mitigate the heteroskedasticity problem. The results of the 2SLS regressions are demonstrated in Table 4.

## Two-stage least squares (2SLS) regressions

Table 5 presents measures showing the suitability of the 2SLS regression model, such as R-square, Durbin Watson, and F-statistic. R-squared is a statistical metric that indicates the extent to which the chosen independent variables explain the fluctuations or changes observed in the dependent variable in a regression model. The Durbin-Watson statistic serves as a test statistic used to identify the presence of autocorrelation in the residuals derived from a regression analysis. The Durbin-Watson statistic ranges between 0 and 4, with a value near 2

**Table 4. Fixed effect regressions.**

| | Model 1 | Model 2 | Model 3 | Model 4 | Model 5 |
|---|---|---|---|---|---|
| Delta SVI | 0.804*** | 1.827*** | 1.826*** | 0.809*** | 0.809*** |
| | (3.83) | (5.98) | (5.98) | (3.46) | (3.46) |
| REV | 0.089*** | -0.086*** | -0.086*** | 0.087*** | 0.087*** |
| | (5.86) | (-4.31) | (-4.35) | (5.25) | (5.25) |
| TURN | 0.045*** | 0.048*** | 0.049*** | 0.051*** | 0.052*** |
| | (4.90) | (7.02) | (7.10) | (5.36) | (5.45) |
| SZ | -0.051 | 1.217*** | 1.213*** | 0.185* | 0.171* |
| | (-0.43) | (3.79) | (3.81) | (1.77) | (1.65) |
| TVOL | 0.281*** | 0.321*** | 0.323*** | 0.292*** | 0.293*** |
| | (7.66) | (7.86) | (7.92) | (11.30) | (11.34) |
| TSKEW | 0.175 | -0.452 | -0.455 | 0.219** | 0.224** |
| | (1.20) | (-1.58) | (-1.59) | (2.19) | (2.24) |
| LnWCC | | 0.271** | | 0.037 | |
| | | (2.01) | | (0.78) | |
| LnWCD | | | 0.346** | | 0.067 |
| | | | (2.37) | | (1.12) |
| Delta SVI_Dummy | | | | 1.052*** | 1.053*** |
| | | | | (2.96) | (2.96) |
| TURN_Dummy | | | | -0.017* | -0.018* |
| | | | | (-1.76) | (-1.91) |
| REV_Dummy | | | | -0.157*** | -0.157*** |
| | | | | (-6.64) | (-6.64) |
| Constant | -0.584 | -32.364*** | -32.255*** | -6.284*** | -6.001** |
| | (-0.21) | (-4.42) | (-4.42) | (-2.65) | (-2.54) |
| Observations | 4,052 | 2,254 | 2,254 | 6,606 | 6,606 |
| R-squared | 0.046 | 0.093 | 0.093 | 0.066 | 0.066 |
| F-statistic | 9.257 | 10.311 | 10.380 | 15.956 | 15.980 |
| Prob(F-statistic) | <0.001 | <0.001 | <0.001 | <0.001 | <0.001 |
| Durbin-Watson stat | 2.026 | 1.954 | 1.954 | 1.994 | 1.995 |
| Hausman Test (Prob.) | <0.001 | <0.001 | <0.001 | <0.001 | <0.001 |
| Redundant Fixed Effect Test (Prob.) | <0.001 | <0.001 | <0.001 | <0.001 | <0.001 |
| Wald Test (Prob.) | <0.001 | <0.001 | <0.001 | <0.001 | <0.001 |

Notes: Table 4 reports the estimations from FEM. All variable definitions are reported in S2 Appendix. T-values are in parentheses. The symbols
***, **, and * denote significance levels of 1%, 5%, and 10%, respectively.

indicating no autocorrelation [55]. The F-test of overall significance indicates whether all independent variables simultaneously affect the dependent variable.

Table 5 presents the 2SLS regression results of five models. Table 5 indicates that reversal significantly impacts crypto returns in pre-pandemic and full-time periods. These findings are similar to those of a prior study by Ozdamar et al. [7], which found a positive correlation between sentiment measures and crypto returns. The implications of these results also align with the overreaction hypothesis in behavioral finance. This hypothesis argues that when investors overreact to news and market events, short-term price overreactions can occur, which are later followed by reversals in the long term. The result also supports Hypothesis 1.

Interestingly, we find a significant positive relationship between investor attention and crypto returns across all periods. Specifically, our findings suggest that an increase in SVI leads

**Table 5. 2SLS estimation results.**

| | Model 1 | Model 2 | Model 3 | Model 4 | Model 5 |
|---|---|---|---|---|---|
| Delta SVI | 0.799*** | 1.889*** | 1.889*** | 0.803*** | 0.803*** |
| | (3.80) | (6.16) | (6.16) | (3.43) | (3.43) |
| REV | 0.099*** | -0.063*** | -0.063*** | 0.093*** | 0.093*** |
| | (6.59) | (-3.20) | (-3.23) | (5.59) | (5.60) |
| TURN | 0.024*** | 0.025*** | 0.026*** | 0.030*** | 0.031*** |
| | (3.08) | (4.94) | (4.99) | (3.52) | (3.57) |
| SZ | 0.039 | 0.201** | 0.205** | 0.136** | 0.136** |
| | (0.58) | (1.98) | (2.03) | (2.39) | (2.39) |
| TVOL | 0.233*** | 0.335*** | 0.338*** | 0.285*** | 0.285*** |
| | (7.11) | (8.88) | (8.94) | (11.66) | (11.69) |
| TSKEW | 0.009 | 0.115 | 0.113 | 0.137* | 0.136* |
| | (0.07) | (0.91) | (0.91) | (1.66) | (1.65) |
| LnWCC | | 0.311** | | 0.073* | |
| | | (2.47) | | (1.82) | |
| LnWCD | | | 0.373*** | | 0.104** |
| | | | (2.69) | | (2.05) |
| Delta SVI_Dummy | | | | 1.096*** | 1.096*** |
| | | | | (3.07) | (3.07) |
| TURN_Dummy | | | | -0.008 | -0.009 |
| | | | | (-0.90) | (-0.98) |
| REV_Dummy | | | | -0.146*** | -0.147*** |
| | | | | (-6.21) | (-6.22) |
| Constant | -2.038 | -8.410*** | -8.341*** | -4.836*** | -4.847*** |
| | (-1.24) | (-3.49) | (-3.48) | (-3.54) | (-3.55) |
| Observations | 4,052 | 2,554 | 2,554 | 6,606 | 6,606 |
| R-squared | 0.039 | 0.076 | 0.077 | 0.057 | 0.057 |
| F-statistic | 27.409 | 29.955 | 30.130 | 39.870 | 39.964 |
| Prob(F-statistic) | <0.001 | <0.001 | <0.001 | <0.001 | <0.001 |
| Durbin-Watson stat | 2.028 | 1.978 | 1.978 | 1.998 | 1.999 |
| Instrument rank | 7 | 8 | 8 | 11 | 11 |

Notes: Table 5 reports the estimations from 2SLS. All variable definitions are reported in S2 Appendix. T-values are in parentheses. The symbols
***, **, and * denote significance levels of 1%, 5%, and 10%, respectively.

to an increase in returns, which aligns with previous research by Subramaniam and Chakraborty [3]. Additionally, our results support the attention-induced price pressure hypothesis proposed by Barber and Odean [13], which suggests that greater investor attention can lead to positive price pressure in the short term and returns reversal in the long term. Moreover, our results indicate that the impact of investor attention on crypto returns during the pandemic is more significant than before the outbreak. This finding suggests that investors were more focused and optimistic about the cryptocurrency market's volatility during the epidemic. Altogether, our results illustrate the importance of investor attention in shaping returns in the cryptocurrency market, particularly during periods of heightened uncertainty and market volatility. This finding also supports hypothesis 2.

Conversely, it was observed that REV and interaction term (REV*Dummy) reduced cryptocurrency returns during the pandemic. Similar conclusions were drawn in the studies conducted by Chen et al. [31], Huo and Qiu [56], supporting the underreaction hypothesis.

Investors may have initially underestimated the impact of the pandemic on the market, indicating an inadequate response to the pandemic's effects on the market. Thus, investors may have undervalued the magnitude and longevity of the pandemic and its influence on the market.

In addition, TURN has a positive correlation with returns, and its volatility was relatively constant across the models. These findings support the conclusions drawn by Dash and Maitra [36] that higher trading activity leads to increased crypto liquidity, which in turn generates higher returns. Additionally, market capitalization positively impacted returns in all periods, which aligns with Subramaniam and Chakraborty [3]. The study provides empirical evidence highlighting the significance of market dynamics, trading activity, and liquidity in shaping cryptocurrency returns.

Furthermore, the number of confirmed COVID-19 cases (WCC) and COVID-19 deaths (WCD) positively strengthen crypto returns. The increased negative news about the epidemic directly impacts investor sentiment, so investors prefer Crypto as a safe haven for their capital. Therefore, the higher demands for cryptos strengthen the crypto returns during the pandemic. Our finding is consistent with Conlon et al. [11]. At the same time, the news of COVID-19 deaths substantially affects investor sentiment more than confirmation of the number of COVID-19 infections. This result aligns with Baig et al. [45].

## How investor attention and reversals affect crypto returns across sub-samples by market capitalization

To better understand the impact of investor attention and reversals on crypto returns across different sub-samples by market capitalization, we followed the robustness test approach proposed by Subramaniam and Chakraborty [3], Hung and Yang [8]. We follow Fama and French [57] to divide the sample into thirds by market capitalization, which ensured an adequate number of observations in each group: small, medium, and large caps, with the smallest 30 percent, the middle 40 percent, and the most significant 30 percent. The regression results in Table 6 illustrate that small and medium cryptocurrencies have more attention from investors than large-cap crypto groups.

Our findings demonstrated that medium-cap investor attention had a 0.16% stronger impact on returns than small-cap cryptos. Furthermore, our analysis revealed that the interaction term, SVI_Dummy, was only statistically significant, with an average coefficient of 1.97 in the medium-cap subsample. This finding indicates that investors preferred investing in medium-cap Crypto during the pandemic. This result aligns with the attention-driven trading behavior theory, which posits that investor attention has predictive power for future returns.

Interestingly, our study found that the reversal positively impacted returns, mainly for small-cap cryptos, which is consistent with the overreact theory of behavioral finance. This result is also similar to Kozlowski et al. [6]. However, we also found that the interaction term of reversal negatively impacted returns. This result aligns with the underreact hypothesis, indicating that investors tend to underreact to the volatility of small-cap cryptos during the pandemic. This result highlights the influence of market sentiment on investor behavior and trading decisions.

We also observed that the number of confirmed COVID-19 cases (WCC) and deaths (WCD) positively impacted medium and large-cap crypto returns, suggesting the pandemic's systemic risk and its effect on market uncertainty. Additionally, other factors such as turnover (TURN), market cap (SZ), total volatility (TVOL), and total skewness (TSKEW) had a more potent effect on medium-cap crypto returns than on small or large-cap cryptos, indicating its higher liquidity due to its moderate market depth. The results show that medium-cap Crypto has more liquidity and reacts to volatility better than small and large-cap cryptos.

**Table 6. Regressions in subsamples by crypto capitalization.**

| | Small-cap | | Medium-cap | | Large-cap | |
|---|---|---|---|---|---|---|
| | **Model 4** | **Model 5** | **Model 4** | **Model 5** | **Model 4** | **Model 5** |
| Delta SVI | 0.778*** | 0.778*** | 0.940** | 0.939*** | -0.093 | -0.092 |
| | (2.56) | (2.56) | (2.03) | (2.03) | (-0.19) | (-0.18) |
| REV | 0.146*** | 0.146*** | -0.017 | -0.017 | -0.013 | -0.013 |
| | (7.08) | (7.08) | (-0.46) | (-0.47) | (-0.39) | (-0.39) |
| TURN | 0.018* | 0.018* | 0.073*** | 0.074*** | 0.029 | 0.029 |
| | (1.86) | (1.88) | (3.42) | (3.51) | (0.94) | (0.96) |
| SZ | -0.037 | -0.036 | 0.683*** | 0.695*** | 0.238 | 0.242 |
| | (-0.40) | (-0.40) | (2.79) | (2.84) | (1.22) | (1.25) |
| TVOL | 0.150*** | 0.150*** | 0.813*** | 0.815*** | 0.216*** | 0.220*** |
| | (5.53) | (5.54) | (11.86) | (11.89) | (3.01) | (3.06) |
| TSKEW | -0.089 | -0.091 | 0.527** | 0.526** | 0.551*** | 0.545*** |
| | (-0.88) | (-0.90) | (2.43) | (2.42) | (3.54) | (3.50) |
| LnWCC | 0.089 | | 0.150* | | 0.151** | |
| | (1.48) | | (1.69) | | (2.38) | |
| LnWCD | | 0.123 | | 0.217* | | 0.195** |
| | | (1.60) | | (1.93) | | (2.48) |
| SVI_Dummy | 0.054 | 0.056 | 1.972*** | 1.969*** | 0.896 | 0.894 |
| | (0.11) | (0.11) | (2.93) | (2.93) | (1.33) | (1.33) |
| TURN_Dummy | -0.013 | -0.013 | -0.028 | -0.030 | 0.002 | 0.002 |
| | (-1.19) | (-1.23) | (-1.28) | (-1.39) | (0.06) | (0.07) |
| REV_Dummy | -0.271*** | -0.272*** | -0.087** | -0.087** | -0.146*** | -0.147*** |
| | (-7.07) | (-7.08) | (-2.00) | (-2.00) | (-3.04) | (-3.05) |
| Constant | 0.380 | 0.363 | -21.989*** | -22.337*** | -7.045 | -7.177 |
| | (0.19) | (0.18) | (-3.71) | (-3.78) | (-1.42) | (-1.46) |
| Observations | 2,956 | 2,956 | 2,104 | 2,104 | 1,546 | 1,546 |
| R-squared | 0.042 | 0.042 | 0.157 | 0.157 | 0.046 | 0.046 |
| F-statistic | 12.825 | 12.864 | 38.901 | 39.006 | 7.351 | 7.403 |
| Prob(F-statistic) | <0.001 | <0.001 | <0.001 | <0.001 | <0.001 | <0.001 |
| Durbin-Watson stat | 1.945 | 1.945 | 1.798 | 1.798 | 1.869 | 1.869 |
| Instrument rank | 11 | 11 | 11 | 11 | 11 | 11 |

Notes: Table 6 reports the robustness test across the subsample size of Crypto using 2SLS estimation. Our results are consistent after a series of robustness tests. All variable definitions are reported in S2 Appendix. T-values are in parentheses. The symbols ***, **, and * denote significance levels of 1%, 5%, and 10%, respectively.

Therefore, our results suggest that understanding investor attention and reversals and considering market capitalization characteristics is crucial in predicting crypto returns during high volatility periods. Our study also contributes to the existing literature by providing insights into the performance of various crypto sub-samples and the impact of COVID-19 on crypto markets.

## How investor attention and reversals affect returns of stablecoins and traditional coins

To investigate the impact of investor attention and reversals on crypto returns, we followed the approach proposed by Wang et al. [52] to divide the sample into stable and traditional coins. Stablecoins are unique because they are pegged to stable and safe assets like gold or

USD, making them more stable and suitable for storing value or providing a safe haven during uncertain times. In contrast, traditional coins are more volatile, as they are not tied to a safe asset like stablecoins. For our purposes, the sample of stablecoins includes USDT and USDC.

Table 7 presents the regression results for stable and traditional coins. The findings indicate that investor attention only positively impacted traditional coin returns. Further analysis showed that the Delta SVI_Dummy indicated a 0.32% stronger impact on traditional coin returns than the Delta SVI variable, implying that investor attention reinforces traditional coin returns during the COVID-19 pandemic. In contrast, the results show that investor attention had no impact on stablecoin returns, suggesting that these coins are less influenced by market sentiment due to their inherent stability. These findings are consistent with the argument of Baur and Hoang [58] that investor attention and bad news increase money flow into traditional coins, leading to higher market volatility and driving capital into stablecoins. Similarly,

**Table 7. Stablecoins versus other coins.**

|  | Stablecoins | | Traditional coins | |
|---|---|---|---|---|
|  | Model 4 | Model 5 | Model 4 | Model 5 |
| Delta SVI | 0.064 | 0.064 | 0.813*** | 0.813*** |
|  | (0.99) | (0.98) | (3.39) | (3.39) |
| REV | -0.337*** | -0.338*** | 0.093*** | 0.093*** |
|  | (-4.11) | (-4.13) | (5.50) | (5.50) |
| TURN | -0.003*** | -0.003*** | 0.035*** | 0.035*** |
|  | (-2.87) | (-2.85) | (3.60) | (3.66) |
| SZ | 0.043 | 0.033 | 0.156*** | 0.155*** |
|  | (0.61) | (0.49) | (2.63) | (2.63) |
| TVOL | 0.041 | 0.033 | 0.285*** | 0.285*** |
|  | (0.37) | (0.30) | (11.21) | (11.23) |
| TSKEW | 0.047 | 0.038 | 0.148* | 0.147* |
|  | (0.59) | (0.49) | (1.76) | (1.75) |
| LnWCC | -0.020 |  | 0.066 |  |
|  | (-0.76) |  | (1.59) |  |
| LnWCD |  | -0.020 |  | 0.096* |
|  |  | (-0.65) |  | (1.82) |
| GSV_Dummy | -0.015 | -0.014 | 1.133*** | 1.133*** |
|  | (-0.16) | (-0.14) | (3.10) | (3.10) |
| TURN_Dummy | 0.003*** | 0.003*** | -0.010 | -0.011 |
|  | (2.65) | (2.63) | (-0.98) | (-1.07) |
| REV_Dummy | 0.311** | 0.312** | -0.148*** | -0.149*** |
|  | (2.44) | (2.45) | (-6.20) | (-6.21) |
| Constant | -0.810 | -0.598 | -5.355*** | -5.364*** |
|  | (-0.57) | (-0.45) | (-3.76) | (-3.76) |
| Observations | 168 | 168 | 6,438 | 6,438 |
| R-squared | 0.246 | 0.245 | 0.058 | 0.058 |
| F-statistic | 5.110 | 5.089 | 39.485 | 39.568 |
| Prob(F-statistic) | <0.001 | <0.001 | <0.001 | <0.001 |
| Durbin-Watson stat | 2.274 | 2.275 | 1.996 | 1.996 |
| Instrument rank | 11 | 11 | 11 | 11 |

Notes: Table 7 reports the robustness test between the stablecoins and other coins using 2SLS estimation. Our results are consistent after a series of robustness tests. All variable definitions are reported in S2 Appendix. T-values are in parentheses. The symbols

***, **, and * denote sae levels of 1%, 5%, and 10%, respectively.

Grobys et al. [59] found that stablecoin returns are less sensitive to macroeconomic factors and more stable than traditional coin returns. In short, the results suggest that stablecoins are a more stable investment option for risk-averse investors. At the same time, traditional coins offer more return opportunities, albeit with higher market volatility. This finding also aligns with the theory of attention-driven trading behavior, which suggests that investor attention has predictive power for future returns.

Additionally, REV hurts the stablecoins returns while positively affecting the traditional coins returns. Besides that, the results show that the COVID-19 pandemic changes the impact of Reversal (REV_Dummy) on both groups. The result aligns with Kozlowski et al. [6], Ozda-mar et al. [7], Eom and Park [9], Bali et al. [27] and supports the underreacted hypothesis. At the same time, TURN showed similar effects with REV in both groups. The results also show that factors such as market cap (SZ), total volatility (TVOL), and total skewness (TSKEW) mainly affect traditional coin returns. In addition, the increase in COVID-19 deaths only leads to increased traditional returns.

In short, the full-time period shows the potential for a positive effect of reversal and turnover on traditional coin returns. Meanwhile, this impact is negative for stablecoins. However, the impact of COVID-19 revealed these effects on both groups. Therefore, the results show a shifting capital flow from traditional to stablecoins during COVID-19. This result is consistent with the argument of Baur and Hoang [58] that investor attention and bad news increase money flow into traditional coins. An increasing market vocality encourages capital to flow into stable coins to hedge risk, implying a significant safe-haven role of stablecoins.

## How investor attention and reversals affect crypto returns during different COVID-19 waves

Table 8 shows the regression results of crypto behavior during each COVID-19 wave. To examine the detailed impact of the COVID-19 pandemic on cryptocurrencies, we follow John Drake's methodology to examine the effect of each wave on crypto returns. He discusses the impact of three COVID-19 waves in the US published in Forbes. The first wave spans from January 22nd, 2020, to May 31st, 2020. The second wave ranges from June 1st, 2020, to August 31st, 2020. The third wave is from September 1st, 2020, to April 15th, 2021.

The results show that investor attention mainly affects returns in the third wave. At the same time, REV negatively affects returns in all three waves. However, the reversal effect tends to decrease with each wave. Besides that, TURN, SZ, and TVOL hurt returns in the first wave. However, TURN, SZ, and TVOL positively affected profits during the second and third waves.

Table 8 reports that the negative impact of COVID-19 decreases with each wave. Specifically, the decrease in the reversal effect shows that investors are less overreacted than in the epidemic's early stages. The proof is that investor attention started to increase in the third wave. At the same time, liquidity and volatility increase during the second and third waves. Moreover, large-cap cryptos tend to be more profitable in the second and third waves than small ones in the first wave. They imply that investors tend to make incorrect decisions in the early stages of the pandemic. However, the long-term market recovery causes the impact of this effect to decrease.

## Conclusion

This study examines how reversals and investor attention affect cryptocurrency returns before and during the pandemic. In addition, we explore the impacts of investors' attention and reversals on crypto returns in different subsamples by market capitalization, stability, and COVID-19 waves. We use data from the top 20 largest capitalization cryptos from January 2nd, 2016, to April 15th, 2021. We employ 2SLS estimation and discuss these results.

**Table 8. Cryptocurrency behavior during different COVID-19 waves.**

|  | First wave | | Second wave | | Third-wave | |
|---|---|---|---|---|---|---|
|  | Model 2 | Model 3 | Model 2 | Model 3 | Model 2 | Model 3 |
| Delta SVI | 0.062 | 0.062 | 0.116 | 0.114 | 3.421*** | 3.417*** |
|  | (0.17) | (0.17) | (0.33) | (0.33) | (6.70) | (6.69) |
| REV | -0.186*** | -0.186*** | -0.153*** | -0.150*** | -0.072*** | -0.073*** |
|  | (-5.10) | (-5.11) | (-3.34) | (-3.28) | (-2.75) | (-2.77) |
| TURN | -0.013** | -0.013** | 0.078*** | 0.077*** | 0.046*** | 0.046*** |
|  | (-2.47) | (-2.45) | (6.58) | (6.47) | (5.40) | (5.43) |
| SZ | -0.418** | -0.413** | 0.327* | 0.349* | 0.429** | 0.397** |
|  | (-2.35) | (-2.32) | (1.77) | (1.89) | (2.30) | (2.12) |
| TVOL | -0.020 | -0.011 | 0.267** | 0.291*** | 0.347*** | 0.352*** |
|  | (-0.22) | (-0.12) | (2.51) | (2.75) | (6.80) | (6.85) |
| TSKEW | -0.443 | -0.437 | 0.236 | 0.258 | 0.078 | 0.023 |
|  | (-1.43) | (-1.41) | (0.89) | (0.97) | (0.39) | (0.11) |
| LnWCC | 0.131 |  | 2.052 |  | 0.791 |  |
|  | (1.13) |  | (1.41) |  | (0.49) |  |
| LnWCD |  | 0.147 |  | 2.647 |  | 2.271 |
|  |  | (1.19) |  | (0.73) |  | (0.96) |
| Constant | 10.032** | 9.987** | -23.127** | -23.891 | -18.167* | -24.297** |
|  | (2.37) | (2.36) | (-2.39) | (-1.29) | (-1.67) | (-1.97) |
| Observations | 599 | 599 | 502 | 502 | 1,453 | 1,453 |
| R-squared | 0.056 | 0.056 | 0.102 | 0.099 | 0.106 | 0.106 |
| F-statistic | 5.015 | 5.034 | 7.972 | 7.740 | 24.375 | 24.483 |
| Prob(F-statistic) | <0.001 | <0.001 | <0.001 | <0.001 | <0.001 | <0.001 |
| Durbin-Watson stat | 1.899 | 1.899 | 1.860 | 1.860 | 1.986 | 1.986 |
| Instrument rank | 8 | 8 | 8 | 8 | 8 | 8 |

Notes: Table 8 reports the robustness test across different pandemic waves using 2SLS estimation. Our results are consistent after a series of robustness tests. All variable definitions are reported in S2 Appendix. T-values are in parentheses. The symbols ***, **, and * denote significance levels of 1%, 5%, and 10%, respectively.

Our findings indicate that reversals significantly impacted crypto returns, with a positive effect before the pandemic in line with the overreaction hypothesis and a negative effect during the pandemic consistent with the underreaction hypothesis. Investor attention positively impacted crypto returns before and during the pandemic, supporting the attention-induced price pressure hypothesis. The number of confirmed COVID-19 cases and deaths increases crypto returns, indicating a safe-haven nature during the pandemic. The study also revealed that investor attention primarily influenced smaller and medium-cap cryptos. At the same time, COVID-19 news positively impacted medium and large-cap cryptos. The negative impact of each wave of COVID-19 was due to market recovery and reduced investor overreaction, and capital flows shifted to stablecoins during the pandemic for risk hedging purposes.

Our findings have important investment implications for individual investors to take advantage of cryptocurrency market reversals. The results suggest that individual investors should estimate their risk preferences before investing in the crypto market. They should understand that the crypto market can be highly volatile, with the potential for significant reversals. This suggestion protects individual investors from the full impact of market reversals. Secondly, implementing stop-loss orders is a proactive strategy to protect investors against sudden reversals. Specifically, the investors need to set stop-loss levels based on their

risk tolerance and market analysis. These orders automatically sell their assets if prices fall to a certain point, helping investors limit losses during market downturns. Finally, investors should stay vigilant and continuously monitor cryptocurrency investments. Market conditions can change rapidly, and reversals may occur without warning. Being adaptable and informed allows investors to make informed decisions to protect their investments during volatile times.

Our research findings hold critical implications for individual investors navigating the cryptocurrency market, particularly during heightened volatility such as the COVID-19 pandemic. Emphasizing the meticulous management of emotions, we advocate for a thorough investigation into the influence of market sentiment and social media on investor decision-making within the cryptocurrency space. Our empirical evidence underscores the substantial impact of investor psychology on small and medium-sized cryptocurrencies, making it imperative for investors to factor this into their strategies. Furthermore, strategic diversification emerges as a paramount consideration. Allocating investments across a spectrum of cryptocurrencies, including stablecoin classes, is an effective risk mitigation strategy. This approach aims to shield investors from the idiosyncratic risks associated with individual assets, fostering resilience in their overall portfolio. These insights provide actionable guidance for investors looking to navigate the nuanced landscape of the cryptocurrency market with prudence and strategic foresight.

In addition, the findings are relevant for policymakers to manage investor sentiment in the cryptocurrency market. Firstly, the governments should empower the regulatory frameworks for the cryptocurrency market. Specifically, governments should work to create clear and comprehensive regulatory frameworks for the cryptocurrency market. This includes defining the legal status of cryptocurrencies, outlining licensing requirements for exchanges and other market participants, and setting standards for investor protection. Moreover, governments may provide regular updates and transparent communication about any changes to regulatory policies. Clarity in regulations helps reduce uncertainty and fosters a more stable investment environment, positively influencing investor sentiment. Finally, the governments may develop financial literacy for investors so they may participate in the market rationally. The governments may collaborate with other countries to establish consistent international standards for the cryptocurrency market. This suggestion can help prevent regulatory arbitrage and create a more cohesive global regulatory environment. Moreover, governments can facilitate global sharing of information and best practices among regulatory bodies. A coordinated effort can enhance the effectiveness of regulatory measures and foster a more secure and trustworthy cryptocurrency market.

While our study extends the growing literature on the cryptocurrency market, it has the following limitations. For instance, analyzing investor attention and reversal impact on the top 20 largest capitalization cryptocurrencies may only partially capture each cryptocurrency's nature. We suggest future research focus on developing a suitable classification system to examine the impact of the characteristics of cryptos on their returns. Additionally, future studies should consider the effect of investor attention on different regions worldwide, as our study mainly focuses on the US market. Lastly, further research on how cryptocurrency becomes a safe-haven asset during the pandemic is required, testing whether sentiment and reversal have a similar impact on other safe-haven assets like bonds or gold. The unusual change in capital flows into the cryptocurrency market also deserves further study.

## Supporting information

**S1 Appendix. Market capitalization and market share of the top 20 largest cryptocurrencies on 1st Apr 2021.**
(DOCX)

**S2 Appendix. Variable definition.**
(DOCX)

**S1 Data.**
(RAR)

## Author Contributions

**Conceptualization:** Huy Pham, Khoa Dang Duong.

**Data curation:** Trang Ngoc Doan Tran.

**Formal analysis:** Ngoc Thi Thanh Nguyen.

**Investigation:** Trang Ngoc Doan Tran, Ngoc Thi Thanh Nguyen.

**Methodology:** Trang Ngoc Doan Tran.

**Project administration:** Huy Pham, Khoa Dang Duong.

**Software:** Trang Ngoc Doan Tran, Ngoc Thi Thanh Nguyen.

**Validation:** Trang Ngoc Doan Tran, Ngoc Thi Thanh Nguyen.

**Writing – original draft:** Huy Pham, Khoa Dang Duong.

**Writing – review & editing:** Huy Pham, Khoa Dang Duong.

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
