## [Decision Letter · Decision Letter 0]

26 Dec 2023

PONE-D-23-39094The Reversal in the Cryptocurrency Market Before and During the COVID-19 Pandemic: Does Investor Attention Matter?PLOS ONE

Dear Dr. Duong,

Thank you for submitting your manuscript to PLOS ONE. After careful consideration, we feel that it has merit but does not fully meet PLOS ONE’s publication criteria as it currently stands. Therefore, we invite you to submit a revised version of the manuscript that addresses the points raised during the review process.

Please address all the comments of the Reviewers' as stated below. Please make sure to strength your motivation, literature review and contribution of the analysis. Please submit your revised manuscript by Feb 09 2024 11:59PM. If you will need more time than this to complete your revisions, please reply to this message or contact the journal office at plosone@plos.org. Please include the following items when submitting your revised manuscript:A rebuttal letter that responds to each point raised by the academic editor and reviewer(s). You should upload this letter as a separate file labeled 'Response to Reviewers'.A marked-up copy of your manuscript that highlights changes made to the original version. You should upload this as a separate file labeled 'Revised Manuscript with Track Changes'.An unmarked version of your revised paper without tracked changes. You should upload this as a separate file labeled 'Manuscript'.

We look forward to receiving your revised manuscript.

Kind regards,

Burcu Kapar

Academic Editor

PLOS ONE

Journal Requirements:

4. In the online submission form, you indicated that "The data underlying the results presented in the study are available from the corresponding author."

Reviewers' comments:

Reviewer's Responses to Questions

**Comments to the Author**

1. Is the manuscript technically sound, and do the data support the conclusions?

Reviewer #1: Yes

Reviewer #2: Yes

2. Has the statistical analysis been performed appropriately and rigorously? 

Reviewer #1: Yes

Reviewer #2: Yes

3. Have the authors made all data underlying the findings in their manuscript fully available?

Reviewer #1: Yes

Reviewer #2: No

4. Is the manuscript presented in an intelligible fashion and written in standard English?

Reviewer #1: Yes

Reviewer #2: Yes

5. Review Comments to the Author

Reviewer #1: Manuscript Number: PONE-D-23-39094

The Reversal in the Cryptocurrency Market Before and During the COVID-19 Pandemic: Does Investor Attention Matter?

Comments to the author

I extend my sincere congratulations to the author(s) for embarking on research regarding such an interesting and important topic. Nevertheless, I have major concerns about the current form of the manuscript. Here are some comments on this matter:

Major Revision

1. The abstract requires revision to make it more attractive. Additionally, there is confusion regarding whether cryptocurrencies or stablecoins serve as a safe haven. The distinction between all cryptocurrencies being stable coins or exhibiting instability is unclear. Try to ensure that the findings are presented cohesively and in a manner that is easily understandable.

2. In the statement “Depending on different market cycles, retail investor attention can pressure asset price prices because …….” it is recommended to review and verify the accuracy of the term 'asset price prices.'

3. The paper's originality is lacking, and the authors should enhance it by providing context on how their study differs from previous research in the field. Specifically, a thorough examination of the studies listed below is recommended, with an emphasis on contextualization and a clear rationale for the current study. It is essential to articulate how the present research contributes to the existing body of knowledge.

https://doi.org/10.1016/j.iref.2021.01.008;

https://doi.org/10.1016/j.techfore.2022.121999;

https://doi.org/10.1080/17520843.2023.2211380

4. In the literature review, the absence of recent studies, specifically those conducted in 2022 and 2023, is noteworthy, as these studies play a crucial role in shaping the formulation of the research hypothesis. Additionally, it is imperative to incorporate studies that focus on the safe haven properties of cryptocurrency in the literature review section. It is essential to include the pivotal studies below to enrich the overall content.

https://doi.org/10.1007/s10690-023-09436-5;

https://doi.org/10.1016/j.frl.2023.103692;

https://doi.org/10.1016/j.jempfin.2022.12.010;

https://doi.org/10.1108/JFEP-02-2023-0047;

https://doi.org/10.1016/j.frl.2022.103360;

https://doi.org/10.1111/ecno.12227;

https://doi.org/10.1007/s10479-023-05517-w;

5. In the data and methodology section, the statement "We collect keyword data every three months to take daily frequency data on Google trends" is unclear and requires additional details regarding the process of selecting keywords for SVI data and the frequency at which SVI data is collected.

6. Introduce the cryptocurrency return calculation process into the methodology section.

7. It is deemed inappropriate that the inclusion of numerous literature pieces pertaining to firm size and turnover on the returns of financial assets on stock returns has an impact on the Estimation Models section. This literature should be appropriately positioned within the literature review section or reduced in the methodology section.

8. The Estimation Models section requires a thorough discussion of the applied methodology. The current version of the Estimation Models section is in need of substantial improvement.

9. The authors should explain how to measure the reversal percentage in the data and methodology section.

10. Incorporate discussions related to model robustness and diagnostic tests into the findings section, demonstrating the consistency of your findings.

11. Also, it is needed to discuss the methodology from which the regression results in Tables 5, 6, and 7 have been derived.

12. It is crucial to emphasise that cryptocurrency represents a highly volatile and risky investment option. Do your findings suggest any implications, and how do they guide the framing of policymaking?

13. Overall, the writing quality and tone are commendable, and the study’s findings are not only interesting but also highly applicable in real-world scenarios.

Reviewer #2: Overall the author has done exciting research by investigating the Cryptocurrency Market Before and During the COVID-19 Pandemic. The authors have conducted an in-depth analysis of the prior research and attempted to address the issues.

Nevertheless, the paper can address the following major issues, before publication.

• In the introduction section the author(s) has to add the motivation part and the findings of the study, which will give a clear understanding of the study. Please have a look and change accordingly.

• The review of the literature section should be systematically arranged. The author(s) has missed some relevant literature on COVID-19 and crypto as suggested. They might add them to the literature.

• Proper justification is required for taking the data period from January 2016 to April 2021.

• The market cap of crypto is changing day by day. How the author measures the market cap, while considering time series of study from 2016 to 2021, justify. and the total market cap of all crypto is shown around 2.09 trillion. Is it point estimation or random?

• As the data set is very long, it is better to check the structure breaks of the data.

• How the author(S) has chosen the small and medium-cap crypto? Is there any criterion?

• How the author(s) has chosen the lead and lag variables?

• The author(s) has used methodological contribution as FE and 2SLS model. What is the justification?

• The sub-sample regression model based on which methodology. Make it clear.

• The author may use a robustness study.

• The author(S) has to add other relevant policy suggestions part into this study.

6. PLOS authors have the option to publish the peer review history of their article (what does this mean?). If published, this will include your full peer review and any attached files.

Reviewer #1: No

Reviewer #2: No

---

## [Author Response · Author response to Decision Letter 0]

5 Feb 2024

We genuinely appreciate anonymous reviewers for constructive feedback. We have revised our paper according to the feedback. We have prepared the following responses to address reviewer questions.

Reviewer #1: Manuscript Number: PONE-D-23-39094

 Comments to the author

 I extend my sincere congratulations to the author(s) for embarking on research regarding such an interesting and important topic. Nevertheless, I have major concerns about the current form of the manuscript. Here are some comments on this matter:

 Major Revision

 1. The abstract requires revision to make it more attractive. Additionally, there is confusion regarding whether cryptocurrencies or stablecoins serve as a safe haven. The distinction between all cryptocurrencies being stable coins or exhibiting instability is unclear. Try to ensure that the findings are presented cohesively and in a manner that is easily understandable. 

Our responses: We thank anonymous reviewers for their recommendations. We have revised our abstract by including more details regarding our findings. 

2. In the statement “Depending on different market cycles, retail investor attention can pressure asset price prices because …….” it is recommended to review and verify the accuracy of the term 'asset price prices.'

Our responses: We thank anonymous reviewers for constructive feedback. We reviewed and edited the term 'asset price prices.' into "can cause volatility." (Page 3, Paragraph 3, highlighted in the manuscript).

 3. The paper's originality is lacking, and the authors should enhance it by providing context on how their study differs from previous research in the field. Specifically, a thorough examination of the studies listed below is recommended, with an emphasis on contextualization and a clear rationale for the current study. It is essential to articulate how the present research contributes to the existing body of knowledge.

https://doi.org/10.1016/j.iref.2021.01.008;

https://doi.org/10.1016/j.techfore.2022.121999;

https://doi.org/10.1080/17520843.2023.2211380

Our responses: We thank the reviewer for the constructive feedback. We clarify the contributions of this study in the introduction section. (Page 5, paragraph 3, highlighted in the manuscript) 

4. In the literature review, the absence of recent studies, specifically those conducted in 2022 and 2023, is noteworthy, as these studies play a crucial role in shaping the formulation of the research hypothesis. Additionally, it is imperative to incorporate studies that focus on the safe haven properties of cryptocurrency in the literature review section. It is essential to include the pivotal studies below to enrich the overall content.

https://doi.org/10.1007/s10690-023-09436-5;

https://doi.org/10.1016/j.frl.2023.103692;

https://doi.org/10.1016/j.jempfin.2022.12.010;

https://doi.org/10.1108/JFEP-02-2023-0047;

https://doi.org/10.1016/j.frl.2022.103360;

https://doi.org/10.1111/ecno.12227;

https://doi.org/10.1007/s10479-023-05517-w;

Our response: Following the reviewer's feedback, we added recent articles that the reviewer recommended focusing on the safe haven properties of cryptocurrency in the literature review section and enriching the overall content. (Sections 2.1 and 2.5, pages 6 and 10, highlighted in the manuscript).

 5. In the data and methodology section, the statement "We collect keyword data every three months to take daily frequency data on Google trends" is unclear and requires additional details regarding selecting keywords for SVI data and the frequency at which SVI data is collected. 

Our response: Following the reviewer's feedback, we added details regarding selecting keywords for SVI data and the frequency at which SVI data is collected. (Page 12, paragraph 1, highlighted in the manuscript).

 6. Introduce the cryptocurrency return calculation process into the methodology section.

Our response: Following the reviewer's feedback, we added the cryptocurrency return calculation process to the methodology section. (Section 3.2, Page 11, highlighted in the manuscript).

 7. It is deemed inappropriate that the inclusion of numerous literature pieces pertaining to firm size and turnover on the returns of financial assets on stock returns has an impact on the Estimation Models section. This literature should be appropriately positioned within the literature review section or reduced in the methodology section.

Our response: We thank the reviewer for the helpful comments. Following the reviewer's feedback, we have moved literature pieces related to size, turnover, and other variables to a separate section and placed them in the literature review section. (Section 2.4, Page 9, highlighted in the manuscript).

 8. The Estimation Models section requires a thorough discussion of the applied methodology. The current version of the Estimation Models section is in need of substantial improvement.

Our response: Following the reviewer's feedback, we have segregated the method section and thoroughly analyzed the employed methodology. (Section 3.5, Page 13,14, highlighted in the manuscript).

 9. The authors should explain how to measure the reversal percentage in the data and methodology section.

Our response: We thank the reviewer for the helpful comment. We have added the reversal measure in the data and methodology section. (Section 3.2, Page 11, highlighted in the manuscript).

10. Incorporate discussions related to model robustness and diagnostic tests into the findings section, demonstrating the consistency of your findings. 

Our response: We thank the reviewer for the helpful comments. We have integrated discussions concerning model robustness and diagnostic tests into the findings section to showcase the consistency of our results. (Paragraph 1, Pages 18, 19 highlighted in the manuscript).

11. Also, it is needed to discuss the methodology from which the regression results in Tables 5, 6, and 7 have been derived.

Our response: We add the methodology from which the regression results in Tables 6, 7, and 8, which is the 2SLS method in the Note section under each table.

 12. It is crucial to emphasise that cryptocurrency represents a highly volatile and risky investment option. Do your findings suggest any implications, and how do they guide the framing of policymaking?

Our response: We added the implications in our conclusion and suggested more implications for investors, managers, and policymakers. (Page 30, paragraph 3 highlighted in the manuscript) 

13. Overall, the writing quality and tone are commendable, and the study’s findings are not only interesting but also highly applicable in real-world scenarios.

Our response: We sincerely thank you for your feedback.

Reviewer #2: Manuscript Number: PONE-D-23-39094

Overall the author has done exciting research by investigating the Cryptocurrency Market Before and During the COVID-19 Pandemic. The authors have conducted an in-depth analysis of the prior research and attempted to address the issues.

Nevertheless, the paper can address the following major issues, before publication.

● In the introduction section the author(s) has to add the motivation part and the findings of the study, which will give a clear understanding of the study. Please have a look and change accordingly. 

Our response: Thank you for your constructive feedback. We add more about our motivation and why we should examine this study in the cryptocurrency market through reality. (Page 4, paragraph 1,2 highlighted in the manuscript).

● The review of the literature section should be systematically arranged. The author(s) has missed some relevant literature on COVID-19 and crypto as suggested. They might add them to the literature.

Our response: Following the reviewer's feedback, we added relevant literature on COVID-19 and crypto that the reviewer suggested, and the literature section was also systematically rearranged in the revised version. (Section 2.5, Page 10, highlighted in the manuscript).

● Proper justification is required for taking the data period from January 2016 to April 2021.

Our response: Thank you for your feedback. We added that proper justification is required for taking the data period from January 2016 to April 2021. (Section 3.1, Page 10,11, highlighted in the manuscript). 

● The market cap of crypto is changing day by day. How the author measures the market cap, while considering time series of study from 2016 to 2021, justify. and the total market cap of all crypto is shown around 2.09 trillion. Is it point estimation or random?

Our response: The total market value of the top 20 cryptocurrencies is around 2 trillion, which was obtained from coinmarketcap.com on April 1st, 2021. We added a footnote for this in our manuscript. (Page 10)

● As the data set is very long, it is better to check the structure breaks of the data.

Our response: We added a panel unit root test using the methodologies of Levin et al. (2002), Im et al. (2003) - IPS, ADF - Fisher, and P.P. - Fisher to check the structure breaks of the data. (Section 4.3, Pages 16, and 17, highlighted in the manuscript).

● How the author(S) has chosen the small and medium-cap crypto? Is there any criterion? 

Our response: We thank the reviewer for the constructive feedback. In our paper, we add and explain how we calculate the small and medium-cap crypto. (Page 22, paragraph 1 in section 4.6, highlighted in the manuscript).

● How the author(s) has chosen the lead and lag variables? 

Our response: We follow Kozlowski et al. (2021) to conduct lag daily return to measure the reversal (REV) in day t, which is the return of the cryptocurrency I over day t-1. (Section 3.2, Page 11).

● The author(s) has used methodological contribution as FE and 2SLS model. What is the justification? 

Our response: We have added an explanation of why we use methodological contribution as FE and 2SLS model (Paragraph 1 in Section 4.4, Page 17, and Methodology section, highlighted in the manuscript). 

● The sub-sample regression model based on which methodology. Make it clear. 

Our response: we justify the sub-sample regression model based on the 2SLS method by making notes under each table (Table 6,7,8).

● The author may use a robustness study.

Our response: Our research has robustness tests and robust results.

● The author(S) has to add other relevant policy suggestions part into this study.

Our response: We added the implications in our conclusion and suggested more implications for policymakers. (Page 30, paragraph 3 highlighted in the manuscript) 

Sahoo, P. K. (2021). COVID-19 pandemic and cryptocurrency markets: an empirical analysis from a linear and nonlinear causal relationship. Studies in Economics and Finance, 38(2), 454-468.

Sahoo, P. K., Sethi, D., & Acharya, D. (2019). Is bitcoin a near stock? Linear and non-linear causal evidence from a price–volume relationship. International Journal of Managerial Finance, 15(4), 533-545.

Sahoo, P. K., & Rath, B. N. (2022). COVID-19 pandemic and bitcoin returns: Evidence from time and frequency domain causality analysis. Asian Economics Letters, 4(Early View).

 REFERENCES

Levin, A., Lin, C. F., & Chu, C. S. J. (2002). Unit root tests in panel data: asymptotic and finite-sample properties. Journal of econometrics, 108(1), 1-24.

Im, K. S., Pesaran, M. H., & Shin, Y. (2003). Testing for unit roots in heterogeneous panels. Journal of econometrics, 115(1), 53-74.

Kozlowski, S. E., Puleo, M. R., & Zhou, J. (2021). Cryptocurrency return reversals. Applied Economics Letters, 28(11), 887-893.

---

## [Decision Letter · Decision Letter 1]

13 May 2024

The Reversal in the Cryptocurrency Market Before and During the COVID-19Pandemic: Does Investor Attention Matter?

PONE-D-23-39094R1

Dear Dr. Duong,

We’re pleased to inform you that your manuscript has been judged scientifically suitable for publication and will be formally accepted for publication once it meets all outstanding technical requirements.

Kind regards,

Joint Chair Prof. Dr. Stelios Bekiros, PhD

Academic Editor

PLOS ONE

Additional Editor Comments (optional):

Reviewers' comments:

The remarks by the expert referees are fully addressed.

Reviewer's Responses to Questions

**Comments to the Author**

1. If the authors have adequately addressed your comments raised in a previous round of review and you feel that this manuscript is now acceptable for publication, you may indicate that here to bypass the “Comments to the Author” section, enter your conflict of interest statement in the “Confidential to Editor” section, and submit your "Accept" recommendation.

Reviewer #1: All comments have been addressed

Reviewer #2: All comments have been addressed

2. Is the manuscript technically sound, and do the data support the conclusions?

Reviewer #1: Yes

Reviewer #2: Yes

3. Has the statistical analysis been performed appropriately and rigorously? 

Reviewer #1: Yes

Reviewer #2: Yes

4. Have the authors made all data underlying the findings in their manuscript fully available?

Reviewer #1: Yes

Reviewer #2: Yes

5. Is the manuscript presented in an intelligible fashion and written in standard English?

Reviewer #1: Yes

Reviewer #2: Yes

6. Review Comments to the Author

Reviewer #1: The authors have duly made the required changes to the manuscript as per the provided suggestions. Overall, the revision is satisfactory.

Reviewer #2: (No Response)

7. PLOS authors have the option to publish the peer review history of their article (what does this mean?). If published, this will include your full peer review and any attached files.

Reviewer #1: No

Reviewer #2: No

---

## [Editor Report · Acceptance letter]

2 Sep 2024

PONE-D-23-39094R1 

PLOS ONE

Dear Dr. Duong, 

I'm pleased to inform you that your manuscript has been deemed suitable for publication in PLOS ONE. Congratulations! Your manuscript is now being handed over to our production team.

Kind regards, 

on behalf of

Professor Dr. Joint Chair Prof. Dr. Stelios Bekiros 

Academic Editor

PLOS ONE